# Association between physical activity and health-related quality of life in middle-aged and elderly individuals with musculoskeletal disorders: Findings from a national cross-sectional study in Korea

**Jung Hyun Lee[1]◉, Il Yun[2,3]◉, Chung-Mo Nam●[4], Suk-Yong Jang[3,5], Eun-Cheol Park●[3,4]***

1 Janssen Korea, Seoul, Republic of Korea, 2 Department of Public Health, Graduate School, Yonsei University, Seoul, Republic of Korea, 3 Institute of Health Services Research, Yonsei University, Seoul, Republic of Korea, 4 Department of Preventive Medicine, Yonsei University College of Medicine, Seoul, Republic of Korea, 5 Department of Healthcare Management, Graduate School of Public Health, Yonsei University, Seoul, Republic of Korea

◉ These authors contributed equally to this work.
* ECPARK@yuhs.ac

**Data Availability Statement:** The KNHANES is publicly asseible. Available online: https://knhanes.kdca.go.kr/.

## Abstract

### Purpose

This study aimed to identify the association between physical activity and health-related quality of life (HRQoL) in middle-aged and elderly individuals with musculoskeletal disorders.

### Methods

This study used data from the 2016–2020 Korea National Health and Nutrition Examination Survey (KNHANES). We included only those over 40 years of age diagnosed with one or more of the following: osteoarthritis, rheumatism, and osteoporosis. In total, 4,731 participants (783 men and 3,948 women) were included as the study population. Multiple logistic regression analysis was performed to examine the association between physical activity and HRQoL.

### Results

In the case of middle-aged and elderly individuals with musculoskeletal disorders, the likelihood of HRQoL worsening was significantly lower for those who regularly engaged in physical activity compared with that of those who did not engage in physical activity at all (men: OR 0.58, 95% CI 0.37–0.90; women: OR 0.64, 95% CI 0.53–0.79). Stratified analysis by the type and intensity of physical activity revealed that the possibility of poor HRQoL was lowest when leisure-related moderate-intensity physical activities were performed (men: OR 0.44, 95% CI 0.22–0.89; Women: OR 0.50, 95% CI 0.36–0.69).

**Funding:** The authors received no specific funding for this work.

**Competing interests:** The authors have declared that no competing interests exist.

## Conclusions

Our findings suggest that engaging in regular physical activity contributes to preventing exacerbation of HRQoL, even if the individual suffers from musculoskeletal disorders. It is necessary to provide an appropriate type and intensity of physical activity in consideration of the patients' pain and severity.

## Introduction

Musculoskeletal disorders are a group of diseases that include osteoarthritis, rheumatoid arthritis, osteoporosis, and back pain, which cause chronic pain in nerves, muscles, and surrounding tissues, such as the neck, shoulder, waist, wrist, and knee. Musculoskeletal disorders severely affect the physical and mental health of patients. The World Health Organization has reported that musculoskeletal disorders are important diseases with a high prevalence worldwide, have a substantial impact on the quality of life and health, and greatly increase the burden of medical expenses [1].

Various genetic and environmental factors affect the occurrence of musculoskeletal disorders [2]. In the current industrialized modern society, simple repetitive movements, improper posture, and lack of exercise cause physical stress. Accumulated physical stress causes repetitive microscopic damage to the muscles, joints, and nerves and weakens the ligaments and tendons, causing chronic pain [3]. Inappropriate health behaviors, such as smoking, drinking, and obesity, are also major causes for the increase in the incidence of musculoskeletal disorders [2].

Joint damage and functional disorders caused by autoimmune diseases, such as rheumatoid arthritis [4], and physical changes due to aging are also major causes of musculoskeletal disorders, which cause pain, fatigue, and physical and psychological deterioration [5]. In particular, musculoskeletal deterioration due to aging, such as a reduction of bone mass and bone density, calcification of tendons and ligaments, and weakening of muscular endurance, progresses rapidly. As a result, walking becomes unstable, and the range of motion of joints and exercise ability decrease [6]. If this process continues chronically, secondary complications, such as cardiovascular disease and metabolic syndrome, are likely to occur, and the quality of life in old age is greatly reduced [7, 8].

Previous studies have revealed that proper physical activity improves physical and mental health and lowers the risk for diseases, such as hypertension, obesity, and heart disease, and improves immune function, which has a positive effect on the quality of life [9–11]. There is a significant difference in the health-related quality of life (HRQoL) of elderly individuals according to age, educational status, sleep time, stress, number of chronic diseases, and subjective health perception [12]. A randomized controlled trial confirmed that those with arthritis experienced higher physical and mental discomfort during the past month than those without arthritis, and those engaging in irregular physical activity had a significantly lower HRQoL than those engaging in recommended levels of physical activity [13]. In patients with rheumatoid arthritis, engaging in regular exercise is emphasized as a necessary measure to improve physical and psychological functions by preventing the deterioration of arthritis and relieving symptoms, such as pain and fatigue [14].

Since the prevalence, severity, and risk of complications of musculoskeletal disorders tend to increase with age, efforts to curb the progression of the disease through appropriate exercise as early as possible are needed. However, most studies on the association between physical

activity and HRQoL in patients with musculoskeletal disorders have been conducted on the elderly, and there have been few studies on patients with musculoskeletal disorders, including osteoarthritis, osteoporosis, and rheumatoid arthritis. Therefore, this study aims to suggest an appropriate level of physical activity by confirming the association between physical activity and HRQoL in middle-aged and older adults with musculoskeletal disorders.

## Materials and methods

### Data and study population

The data used in this study was obtained from the 2016–2020 Korea National Health and Nutrition Examination Survey (KNHANES), a cross-sectional and nationwide survey conducted by the Korea Disease Control and Prevention Agency (KDCA). KNHANES was designed to evaluate the health status, health behavior, and nutritional status in South Korea in order to provide basic data for developing nationwide health policies [15, 16]. Informed consent was obtained from all respondents in advance, and all data analyzed in this study were completely anonymized. Since KNHANES complies with the Declaration of Helsinki and provides publicly accessible data, no further ethical approval was required for the use of this data [16, 17].

The total survey population from the recent five years (2016–2020) included 39,738 individuals. Among these individuals, we only included middle-aged and elderly individuals over 40 years of age who were diagnosed with one or more of the following conditions in the analysis: osteoarthritis, rheumatism, and osteoporosis. Therefore, those under 40 years of age ($N$ = 16,483) and those without musculoskeletal diseases ($N$ = 18,483) were excluded. Finally, after excluding missing data ($N$ = 41), 4,731 participants (783 men; 3,948 women) constituted the study sample.

### Measures

The dependent variable was HRQoL, which was defined using the European Quality of Life-5 Dimensions (EQ-5D). EQ-5D is a tool developed by EuroQol, which evaluates five dimensions (exercise ability, self-care, daily activities, pain/discomfort, and anxiety/depression) on a 3-point scale, where "1" indicates no disturbance, "2" indicates moderate disturbance, and "3" indicates severe disturbance [18]. We calculated the EQ-5D index according to the Korean EQ-5D measurement criteria presented by KDCA and obtained the quality weight of the index using an arithmetic formula [19]. A response of "1" for all five dimensions indicates the best state of health, and the EQ-D value in this case is defined as 1. On the other hand, if there is a response of "2" or "3", EQ-5D is calculated as 1 –quality of weight [20]. The worst state of health is indicated when the responses of all five dimensions are "3", and the EQ-5D value in this case is -0.171. Therefore, the EQ-5D index has a value of -0.0171 to 1. That is, the smaller the EQ-5D value, the worse the health status. Based on the average of the EQ-5D index, the group with an above-average EQ-5D index was defined as the group with a good HRQoL, and the group with a below-average EQ-5D index was defined as the group with a poor HRQoL.

The variable of interest was physical activity, which was classified into three groups: the no physical activity group, the regular physical activity group, and the irregular physical activity group. The level of physical activity was defined with the help of the Metabolic Equivalent of Task (MET) value according to the intensity of physical activity using the Global Physical Activity Questionnaire (GPAQ) [21]. MET is an indication of physical activity intensity and is a value that specifies the amount of oxygen required to maintain a stable state of 1 MET. It expresses the amount of oxygen consumed during various activities as a multiple of 1 MET. High-intensity activity is vigorous physical activity that results in breathlessness or an

increased heart rate; it is rated at 8.0 METs. Moderate-intensity activity refers to moderate physical activity that results in slight shortness of breath or slightly increased heart rate; it is rated at 4.0 METs. The amount of weekly physical activity (METs/w) is calculated by multiplying the number of days and the METs level by each minute of work-related activities, activities while moving from place to place, and leisure activities that are usually performed for more than 10 minutes during the week [22]. According to the recommended level of physical activity practice for Koreans presented by KDCA, the criteria for regular physical activity were defined as follows: 1) The cumulative amount of physical activity performed regularly per week is 600 METs/w, or 2) Moderate-intensity physical activity performed for 150 minutes or more per week, or 3) 75 minutes or more of high-intensity physical activity per week [23, 24]. Accordingly, in our study, those who met the above criteria were defined as the regular physical activity group. On the other hand, those who could not calculate METs by engaging in physical activity for less than 10 minutes in the last week were classified as the no physical activity group, and those who did physical activity but not meet the above criteria were classified as the irregular physical activity group.

The covariates included demographic factors (sex, age, marital status, and educational level), socioeconomic factors (income, region, and economic activity), and health-behavior patterns (drinking and smoking). Additionally, to control the severity of musculoskeletal disorders, we adjusted the variable related to activity limitation due to arthritis, fractures, joint injuries, and types of musculoskeletal disorders.

## Statistical analysis

The chi-squared test was used to investigate and compare the general characteristics of the study population. Multiple logistic regression analysis was conducted subsequently to investigate the association between physical activity and HRQoL among middle-aged and elderly individuals with musculoskeletal disorders. The odds ratios (ORs) and 95% confidence intervals (CI) were presented as the key results. SAS version 9.4 (SAS Institute Inc; Cary, NC, USA) was used for all analyses, and a p-value of <0.05 was considered statistically significant.

## Results

Table 1 presents the general characteristics of the study population separated by sex. Among the total study sample(N = 4,731), 783 (16.6%) were men, and 3,948 (86.4%) were women. Women were affected more frequently by musculoskeletal disorders. Moreover, 40.0% of men and 34.2% of women engaged in regular physical activity, whereas 42.9% of men and 42.1% of women did not engage in physical activity at all. The average EQ-5D index for the participants in this study was 0.861 for men and 0.849 for women. Therefore, 66.5% of men and 65.6 of women had EQ-5D values above average and good HRQoL.

Table 2 demonstrates the adjusted associations of physical activity with HRQoL. Compared with those who did not engage in physical activity at all, those who regularly engaged in physical activity were significantly less likely to have below-average EQ-5D (men: OR 0.58, 95% CI 0.37–0.90; women: OR 0.64, 95% CI 0.53–0.79).

As shown in Table 3, subgroup analysis stratified by the type and intensity of physical activity was also performed. When stratified by the type of physical activity, the association between leisure-related activity and HRQoL was the greatest (men: OR 0.49, 95% CI 0.26–0.94; women: OR 0.49, 95% CI 0.36–0.66). When stratified by the intensity of physical activity, physical activity of moderate intensity had the greatest effect on HRQoL (men: OR 0.95, 95% CI 0.45–1.03; women: OR 0.69, 95% CI 0.58–0.82). Lastly, when considering both the type and intensity of physical activity, the possibility of poor HRQoL was lowest for men when they engaged in

**Table 1. General characteristics of the study population.**

| Variables | Men | | | | | | Women | | | | | |
|---|---|---|---|---|---|---|---|---|---|---|---|---|
| | Health-related quality of life (HRQoL) | | | | | | Health-related quality of life (HRQoL) | | | | | |
| | TOTAL | | Good (EQ-5D score ≥ Mean*) | | Poor (EQ-5D score < Mean*) | | P-value | TOTAL | | Good (EQ-5D score ≥ Mean*) | | Poor (EQ-5D score < Mean*) | | P-value |
| | N | % | N | % | N | % | | N | % | N | % | N | % | |
| Total (N = 4,731) | 783 | 100.0 | 521 | 66.5 | 262 | 33.5 | | 3,948 | 100.0 | 2,589 | 65.6 | 1,359 | 34.4 | |
| **Physical activity** | | | | | | | < .0001 | | | | | | | < .0001 |
| None | 336 | 42.9 | 200 | 59.5 | 136 | 40.5 | | 1,663 | 42.1 | 956 | 57.5 | 707 | 42.5 | |
| Irregular practice group | 134 | 17.1 | 83 | 61.9 | 51 | 38.1 | | 934 | 23.7 | 634 | 67.9 | 300 | 32.1 | |
| Regular practice group | 313 | 40.0 | 238 | 76.0 | 75 | 24.0 | | 1,351 | 34.2 | 999 | 73.9 | 352 | 26.1 | |
| **Age** | | | | | | | < .0001 | | | | | | | < .0001 |
| 40~49 | 32 | 4.1 | 27 | 84.4 | 5 | 15.6 | | 148 | 3.7 | 130 | 87.8 | 18 | 12.2 | |
| 50~59 | 127 | 16.2 | 105 | 82.7 | 22 | 17.3 | | 650 | 16.5 | 533 | 82.0 | 117 | 18.0 | |
| 60~69 | 252 | 32.2 | 174 | 69.0 | 78 | 31.0 | | 1,370 | 34.7 | 970 | 70.8 | 400 | 29.2 | |
| over 70 | 372 | 47.5 | 215 | 57.8 | 157 | 42.2 | | 1,780 | 45.1 | 956 | 53.7 | 824 | 46.3 | |
| **Marital status** | | | | | | | 0.0003 | | | | | | | < .0001 |
| Married or Cohabiting | 661 | 84.4 | 457 | 69.1 | 204 | 30.9 | | 2,385 | 60.4 | 1,737 | 72.8 | 648 | 27.2 | |
| Else | 122 | 15.6 | 64 | 52.5 | 58 | 47.5 | | 1,563 | 39.6 | 852 | 54.5 | 711 | 45.5 | |
| **Income level** | | | | | | | < .0001 | | | | | | | < .0001 |
| Low | 290 | 37.0 | 150 | 51.7 | 140 | 48.3 | | 1,582 | 40.1 | 847 | 53.5 | 735 | 46.5 | |
| Middle | 363 | 46.4 | 263 | 72.5 | 100 | 27.5 | | 1,783 | 45.2 | 1,271 | 71.3 | 512 | 28.7 | |
| High | 130 | 16.6 | 108 | 83.1 | 22 | 16.9 | | 583 | 14.8 | 471 | 80.8 | 112 | 19.2 | |
| **Educational level** | | | | | | | < .0001 | | | | | | | < .0001 |
| Low | 249 | 31.8 | 136 | 54.6 | 113 | 45.4 | | 2,023 | 51.2 | 1,120 | 55.4 | 903 | 44.6 | |
| Middle | 373 | 47.6 | 252 | 67.6 | 121 | 32.4 | | 1,531 | 38.8 | 1,135 | 74.1 | 396 | 25.9 | |
| High | 161 | 20.6 | 133 | 82.6 | 28 | 17.4 | | 394 | 10.0 | 334 | 84.8 | 60 | 15.2 | |
| **Region** | | | | | | | 0.099 | | | | | | | < .0001 |
| Urban area | 331 | 42.3 | 231 | 69.8 | 100 | 30.2 | | 1,766 | 44.7 | 1,227 | 69.5 | 539 | 30.5 | |
| Rural area | 452 | 57.7 | 290 | 64.2 | 162 | 35.8 | | 2,182 | 55.3 | 1,362 | 62.4 | 820 | 37.6 | |
| **Economic activity** | | | | | | | < .0001 | | | | | | | < .0001 |
| Yes | 404 | 51.6 | 300 | 74.3 | 104 | 25.7 | | 1,475 | 37.4 | 1,095 | 74.2 | 380 | 25.8 | |
| No | 379 | 48.4 | 221 | 58.3 | 158 | 41.7 | | 2,473 | 62.6 | 1,494 | 60.4 | 979 | 39.6 | |
| **Activity limitation due to arthritis, fractures, and joint injuries** | | | | | | | < .0001 | | | | | | | < .0001 |
| Yes | 79 | 10.1 | 19 | 24.1 | 60 | 75.9 | | 364 | 9.2 | 87 | 23.9 | 277 | 76.1 | |
| No | 704 | 89.9 | 502 | 71.3 | 202 | 28.7 | | 3,584 | 90.8 | 2,502 | 69.8 | 1,082 | 30.2 | |
| **Drinking** | | | | | | | 0.002 | | | | | | | < .0001 |
| Yes | 544 | 69.5 | 381 | 70.0 | 163 | 30.0 | | 1,755 | 44.5 | 1,272 | 72.5 | 483 | 27.5 | |
| No | 239 | 30.5 | 140 | 58.6 | 99 | 41.4 | | 2,193 | 55.5 | 1,317 | 60.1 | 876 | 39.9 | |
| **Smoking** | | | | | | | 0.211 | | | | | | | 0.0004 |
| Yes | 168 | 21.5 | 105 | 62.5 | 63 | 37.5 | | 119 | 3.0 | 60 | 50.4 | 59 | 49.6 | |
| No | 615 | 78.5 | 416 | 67.6 | 199 | 32.4 | | 3,829 | 97.0 | 2,529 | 66.0 | 1,300 | 34.0 | |
| **Type of musculoskeletal disease** | | | | | | | 0.060 | | | | | | | < .0001 |
| Osteoarthritis | 588 | 75.1 | 378 | 64.3 | 210 | 35.7 | | 2,604 | 66.0 | 1,578 | 60.6 | 1,026 | 39.4 | |
| Rheumatism | 84 | 10.7 | 60 | 71.4 | 24 | 28.6 | | 264 | 6.7 | 192 | 72.7 | 72 | 27.3 | |
| Osteoporosis | 111 | 14.2 | 83 | 74.8 | 28 | 25.2 | | 1,080 | 27.4 | 819 | 75.8 | 261 | 24.2 | |
| **Year** | | | | | | | 0.362 | | | | | | | 0.012 |
| 2016 | 137 | 17.5 | 88 | 64.2 | 49 | 35.8 | | 783 | 19.8 | 496 | 63.3 | 287 | 36.7 | |
| 2017 | 171 | 21.8 | 110 | 64.3 | 61 | 35.7 | | 833 | 21.1 | 555 | 66.6 | 278 | 33.4 | |
| 2018 | 161 | 20.6 | 108 | 67.1 | 53 | 32.9 | | 815 | 20.6 | 510 | 62.6 | 305 | 37.4 | |
| 2019 | 148 | 18.9 | 94 | 63.5 | 54 | 36.5 | | 816 | 20.7 | 534 | 65.4 | 282 | 34.6 | |
| 2020 | 166 | 21.2 | 121 | 72.9 | 45 | 27.1 | | 701 | 17.8 | 494 | 70.5 | 207 | 29.5 | |

* The mean of the EQ-5D index (Men: 0.861; Women: 0.849)

**Table 2. Results of factors associated with Health-related quality of life (HRQoL).**

| Variables | | Men | | | | Women | | | |
|---|---|---|---|---|---|---|---|---|---|
| | | EQ-5D score < Mean* | | | | EQ-5D score < Mean* | | | |
| | | OR | 95% CI | | | OR | 95% CI | | |
| **Physical activity** | | | | | | | | | |
| | None | 1.00 | | | | 1.00 | | | |
| | Irregular practice group | 0.99 | (0.58 | - | 1.68) | 0.77 | (0.62 | - | 0.96) |
| | Regular practice group | 0.58 | (0.37 | - | 0.90) | 0.64 | (0.53 | - | 0.79) |

* The mean of the EQ-5D index (Men: 0.861; Women: 0.849)

leisure-related moderate-intensity physical activities (OR 0.44, 95% CI 0.22–0.89), whereas it was the lowest for women they engaged in leisure-related high-intensity physical activities (OR 0.41, 95% CI 0.17–0.97).

Additional Table 1 shows the results of the subgroup analysis according to the five components of the EQ-5D index. In the case of men with musculoskeletal disorders, exercise ability, self-care, and daily activity were confirmed to have improved when physical activities were performed regularly. In contrast, women showed significant associations among all four components except for anxiety/depression.

Additional sensitivity analyses were conducted to compare the results when different criteria were applied for defining HRQoL. As noted in Additional Table 2, the cut-offs of the EQ-5D score were applied as the 25th percentiles, median, and 75th percentiles. However, regardless of the criterion applied, it was confirmed that the possibility of poor HRQoL was lowered when physical activity was performed regularly.

**Table 3. Results of subgroup analysis stratified by the type and intensity of physical activity.**

| Variables | Male | | | | Female | | | |
|---|---|---|---|---|---|---|---|---|
| | EQ-5D score < Mean* | | | | EQ-5D score < Mean* | | | |
| | OR | 95% CI | | | OR | 95% CI | | |
| **Type of physical activity** | | | | | | | | |
| None | 1.00 | | | | 1.00 | | | |
| Work-related activities | 1.25 | (0.56 | - | 2.81) | 1.88 | (1.25 | - | 2.83) |
| Walking | 0.72 | (0.46 | - | 1.12) | 0.68 | (0.57 | - | 0.82) |
| Leisure-related activities | 0.49 | (0.26 | - | 0.94) | 0.49 | (0.36 | - | 0.66) |
| **Intensity of physical activity** | | | | | | | | |
| None | 1.00 | | | | 1.00 | | | |
| High | 0.81 | (0.31 | - | 2.11) | 0.94 | (0.54 | - | 1.65) |
| Moderate | 0.68 | (0.45 | - | 1.03) | 0.69 | (0.58 | - | 0.82) |
| **Type and intensity of physical activity** | | | | | | | | |
| None | 1.00 | | | | 1.00 | | | |
| Work-related high-intensity activities | 0.94 | (0.25 | - | 3.52) | 4.71 | (1.71 | - | 13.01) |
| Work-related moderate-intensity activities | 1.38 | (0.54 | - | 3.55) | 1.62 | (1.05 | - | 2.50) |
| Moderate-intensity walking | 0.72 | (0.46 | - | 1.12) | 0.68 | (0.57 | - | 0.82) |
| Leisure-related high-intensity activities | 0.76 | (0.22 | - | 2.56) | 0.41 | (0.17 | - | 0.97) |
| Leisure-related moderate-intensity activities | 0.44 | (0.22 | - | 0.89) | 0.50 | (0.36 | - | 0.69) |

* The mean of the EQ-5D index (Men: 0.861; Women: 0.849)

## Discussion and conclusions

This study explored the association between physical activity and HRQoL in middle-aged and elderly patients with musculoskeletal disorders based on representative national data and valid indicators. The key findings were: 1) Even with musculoskeletal disorders, regular physical activity had a significant effect on their HRQoL, and 2) Leisure-related moderate-intensity physical activity had the greatest effect on preventing exacerbation of HRQoL.

Our findings are consistent with the existing evidence that lack of physical activity and activity limitation are fetal physical and psychological risk factors for the management of osteoarthritis and are associated with HRQoL. Several previous studies demonstrated that interventions related to exercise and weight control are effective in controlling symptoms and maintaining quality of life in patients with osteoarthritis [25, 26]. In addition, a domestic cross-sectional study analyzed the data from the 2016–2017 KNHANES and reported the association between aerobic physical activity and HRQoL in women with osteoarthritis aged 40–59 years. They found that middle-aged women with osteoarthritis had a significantly poorer HRQoL than that of the general population; however, the aerobic physical activity group had a better HRQoL than that of the non-practicing group [27]. Another previous study explored the effects of 6 weeks of short-term, high-intensity exercise on pain, joint function, and quality of life in patients with knee osteoarthritis aged 36–65 years. They reported that the planned 6-week exercise had no significant effect on the improvement of pain and joint function but had a positive effect on the patients' quality of life [28]. Their findings, which demonstrated that physical activity positively affects HRQoL in patients with musculoskeletal disorders, are consistent with our findings. Although they assessed only one type or intensity of exercise, we identified the most effective combination of exercise type and intensity to enhance patients' HRQoL and suggested practical applications.

Except for a few studies, most previous studies on the relationship between physical activity and HRQoL were conducted on older patients [29–31]. In addition, few studies targeted patients with musculoskeletal disorders, including osteoporosis, rheumatoid arthritis, and osteoarthritis. According to the age distribution of our study population, middle-aged patients account for almost 50%, indicating that musculoskeletal disorders are no longer diseases of the elderly. In addition, rheumatism and osteoporosis also showed a fairly high prevalence. Therefore, our study is meaningful in that it clearly grasps the current prevalence of the disease in Korea and selects the target group and target disease for the study.

Our study had certain limitations. First, as we conducted analyses with a cross-sectional survey, only the associations could be confirmed, whereas causality could not be confirmed. In addition, the data was self-reported; hence, the frequency and intensity of physical activity that individuals actually engaged in may not have been accurately measured and was less reliable. Second, although we tried to correct for various covariates that may affect the dependent variable, residual confounding from unmeasured variables could not be ruled out. Third, as our study focused on individuals diagnosed with musculoskeletal disorders, patients who were unable to engage in regular exercise due to advanced musculoskeletal disorders were also included in the analysis. Therefore, the size of association may have been underestimated. Due to the nature of the cross-sectional study, inverse associations are also possible, so the results should be interpreted cautiously. Despite the limitations of the data used, our study had a great advantage in that it evaluated the participants' HRQoL and physical activity using the validated indicators EQ-5D and MET. Moreover, we tried to compensate for the limitation of not determining the severity by correcting the type of musculoskeletal disease and the activity limitation due to arthritis, fractures, or joint injuries. It is meaningful in that we suggested the optimal type and intensity of physical activity to improve HRQoL in middle-aged and elderly patients with musculoskeletal disorders through stratified analysis.

In conclusion, our findings identified that engaging in regular physical activity contributes to preventing exacerbation of HRQoL, even if the individual suffers from musculoskeletal disorders. Especially noteworthy was the significant association found between leisure-related moderate-intensity activity and high HRQoL, suggesting that a leisurely exercise program that musculoskeletal patients can enjoy should be provided to improve their health-related quality of life. It should be provided manuals of the appropriate type and intensity in consideration of the patients' pain and severity. Furthermore, interventions to improve physical activity need to be introduced based on smartphones and wearable devices to increase accessibility, and education on manuals is needed to increase participation in interventions [32].

## Acknowledgments

We would like to thank the members of the Institute of Health Services Research at Yonsei University for their advice for the further development of this study.

## Author Contributions

**Conceptualization:** Jung Hyun Lee.

**Data curation:** Jung Hyun Lee, Il Yun, Chung-Mo Nam, Suk-Yong Jang.

**Formal analysis:** Il Yun.

**Investigation:** Jung Hyun Lee, Il Yun, Suk-Yong Jang, Eun-Cheol Park.

**Methodology:** Il Yun, Chung-Mo Nam, Suk-Yong Jang, Eun-Cheol Park.

**Software:** Chung-Mo Nam.

**Supervision:** Eun-Cheol Park.

**Validation:** Suk-Yong Jang, Eun-Cheol Park.

**Writing – original draft:** Jung Hyun Lee, Il Yun.

**Writing – review & editing:** Jung Hyun Lee, Il Yun.

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
