## [Decision Letter · Decision Letter 0]

9 Aug 2023

PONE-D-23-15070Association between physical activity and health-related quality of life in middle-aged and elderly individuals with musculoskeletal disorders: Findings from a national cross-sectional study in KoreaPLOS ONE

Dear Dr. Park,

Thank you for submitting your manuscript to PLOS ONE. After careful consideration, we feel that it has merit but does not fully meet PLOS ONE’s publication criteria as it currently stands. Therefore, we invite you to submit a revised version of the manuscript that addresses the points raised during the review process.

ACADEMIC EDITOR:

Dear Authors,

The reviewers completed the assessment and have provided constructive feedback and suggestions for further improvement of your manuscript. Their insights aim to enhance the quality and impact of your research. We believe that addressing their comments will significantly strengthen your work and ensure its suitability for publication in Heliyon.

Based on the reviewers' comments, we request you carefully revise your manuscript. We understand that this process requires time and effort, but we firmly believe that your commitment to refining the work will result in a more robust and impactful publication.

We have included the reviewers' comments and suggestions below to assist you in the revision process. Please review them attentively and address each comment in your revised manuscript. It is crucial to provide clear, concise, and well-supported responses to the reviewers' concerns and incorporate the necessary changes in your manuscript.

Additionally, please make sure to indicate the modifications you have made, point-by-point, in a response letter. This will facilitate the identification of the revisions during the subsequent review process.

We look forward to receiving your revised manuscript.

Kind regards,

Ragab K. Elnaggar

Academic Editor

PLOS ONE

Journal Requirements:

Reviewers' comments:

Reviewer's Responses to Questions

**Comments to the Author**

1. Is the manuscript technically sound, and do the data support the conclusions?

Reviewer #1: Partly

2. Has the statistical analysis been performed appropriately and rigorously? 

Reviewer #1: No

3. Have the authors made all data underlying the findings in their manuscript fully available?

Reviewer #1: Yes

4. Is the manuscript presented in an intelligible fashion and written in standard English?

Reviewer #1: Yes

5. Review Comments to the Author

Reviewer #1: There is mistake in total number of participants in the Abstract and Data and study population section.

The conclusions in Abstract are not supported by study results. Based on cross-sectional nature of research it is not possible to conclude that physical activity can improve HRQoL. It should be corrected in the whole manuscript.

In Measures section, lines 110 – 111 should be placed in Results section.

In lines 111-114 is explained the procedure of categorizing participants into groups with poor HRQoL and good HRQoL. It is unclear why participants with “below-average EQ-5D index” are categorized as “poor HRQoL” if “A response of “1” for all five dimensions indicates the best state of health…”. Explanation of scoring protocol indicate that lower result is better result. Please explain and/or correct procedure of categorizing participants into 2 groups according to their HRQoL.

Line 115: “The variable of interest was physical exercise…” This is not correct because GPAQ was used to assess total physical activity not exercise. Please correct.

In Measures section it is not explained how participants were categorized into 3 groups according to their physical activity level. It is only explained cut-off point for “regular physical activity”. Please add explanation of cut-off points for other two groups.

Line 147: number of participants should be corrected in the whole text. The number of participants is differently presented in the Abstract and Data and study population section.

In Table 1 it is written “Bad EQ-5D score” although in Methods section of the manuscript is was called “poor EQ-5D score”. Please correct.

In Table 2 all variables, except for physical activity, are irrelevant for the presented study. I think it should be excluded from the Table 2.

In Table 3 is presented subgroup analysis stratified by the type and intensity of physical activity but it is not clear how participants were categorized in subgroups of physical activity. What were the cut-off points? How were the groups defined? It does not make sense to compare groups if some of participants can belong to more than one group. Please explain how you categorized participants and explain the meaning of this results.

Additional table 1 and 2 are missing from the manuscript.

In Discussion section results are put in the context of very few previous studies neither without explanation of possible reasons for determined association nor possible implications of findings in terms of public health interventions and recommendations for physical activity programs tailored to the identified population

6. PLOS authors have the option to publish the peer review history of their article (what does this mean?). If published, this will include your full peer review and any attached files.

Reviewer #1: No

---

## [Author Response · Author response to Decision Letter 0]

3 Sep 2023

We thank you for giving us the opportunity to revise our paper. In revising our paper, we have carefully considered the comments and suggestions of editor and reviewers, and have done our best to incorporate them accordingly. As instructed, we have attempted to explain the changes made in reaction to all of the reviewer’s comments. The reviewer’s comments were very helpful overall, and we appreciate the constructive feedback on our submission. After addressing the issues raised, we feel the quality of the paper has greatly improved and we hope you agree. We attach a revised manuscript with the track changes and point-by-point response note. Again, thank you for the valuable and helpful comments.

---

## [Editor Report · Decision Letter 1]

6 Oct 2023

PONE-D-23-15070R1Association between physical activity and health-related quality of life in middle-aged and elderly individuals with musculoskeletal disorders: Findings from a national cross-sectional study in KoreaPLOS ONE

Dear Dr. Park,

Thank you for submitting your manuscript to PLOS ONE. After careful consideration, we feel that it has merit but does not fully meet PLOS ONE’s publication criteria as it currently stands. Therefore, we invite you to submit a revised version of the manuscript that addresses the points raised during the review process.

The authors have responded to the previous reviewer's comments and used these comments to revised their work. However, there are still some problems that need to be fixed.1. It seems that the authors uploaded the original version as the clean version for the resubmission. Specifically, the numbers are still wrong in the Abstract for the clean version, although the tracked change version has corrected the typos. The authors should recheck again regarding their data accuracy and the uploaded materials.2. The authors still have some unclear information regarding the data. For example, The authors said, "The total survey population from the recent five years (2016–2020) included 39,738 individuals. Among these individuals, we only included middle-aged and elderly individuals over 40 years of age who were diagnosed with one or more of the following conditions in the analysis: osteoarthritis, rheumatism, and osteoporosis. After excluding missing data (N=41), 4,731 participants (783 men; 3,948 women) constituted the study sample."It is unclear how many participants were available for those age over 40 years. From 39,738 to 4,731 is large reduction. Therefore, it has to be clear to indicate the data cleaning process. It will be even better if the authors could have a flow chart to indicate how they remove the participants from the 39,738 to the final 4,731.3. Following the previous comment, the present sample includes a large proportion of female participants. Is this the case for Korean's gender distribution fro middle-aged and elderly individuals with musculoskeletal disorders? 4. When mentioning the interventions on physical activity improvement, please also indicate the potential programs on education and smartphone use. Please see the following papers. Joveini H, Malaijerdi Z, Sharifi N, Borghabani R, Hashemian M. A theory-based educational intervention to promote behavior change and physical activity participation in middle-aged women: A randomized controlled trial. Asian J Soc Health Behav 2022;5:93-100Huang PC, Chen JS, Potenza MN, et al. Temporal associations between physical activity and three types of problematic use of the internet: A six-month longitudinal study. J Behav Addict. 2022;11(4):1055-1067.Saffari M, Chen JS, Wu HC, et al. Effects of Weight-Related Self-Stigma and Smartphone Addiction on Female University Students' Physical Activity Levels. Int J Environ Res Public Health. 2022;19(5):2631.Xu P, Chen JS, Chang YL, et al. Gender Differences in the Associations Between Physical Activity, Smartphone Use, and Weight Stigma. Front Public Health. 2022;10:862829. 

We look forward to receiving your revised manuscript.

Kind regards,

Chung-Ying Lin

Academic Editor

PLOS ONE

---

## [Author Response · Author response to Decision Letter 1]

31 Oct 2023

Thank you for the valuable and helpful comments. We attached a revised manuscript with the track changes, clean version manuscript file, and our response to each point.

---

## [Editor Report · Decision Letter 2]

5 Nov 2023

Association between physical activity and health-related quality of life in middle-aged and elderly individuals with musculoskeletal disorders: Findings from a national cross-sectional study in Korea

PONE-D-23-15070R2

Dear Dr. Park,

We’re pleased to inform you that your manuscript has been judged scientifically suitable for publication and will be formally accepted for publication once it meets all outstanding technical requirements.

Kind regards,

Chung-Ying Lin

Academic Editor

PLOS ONE

Additional Editor Comments (optional):

The authors have satisfactorily addressed the reviewer's comments. I think that the paper is publishable in the present form. 
---

## [Editor Report · Acceptance letter]

8 Nov 2023

PONE-D-23-15070R2 

Association between physical activity and health-related quality of life in middle-aged and elderly individuals with musculoskeletal disorders: Findings from a national cross-sectional study in Korea 

Dear Dr. Park:

I'm pleased to inform you that your manuscript has been deemed suitable for publication in PLOS ONE. Congratulations! Your manuscript is now with our production department. 

Kind regards, 

on behalf of

Dr. Chung-Ying Lin 

Academic Editor

PLOS ONE